# Intranasal Administration of a TRAIL Neutralizing Monoclonal Antibody Adsorbed in PLGA Nanoparticles and NLC Nanosystems: An In Vivo Study on a Mouse Model of Alzheimer’s Disease

**DOI:** 10.3390/biomedicines10050985

**Published:** 2022-04-23

**Authors:** Teresa Musumeci, Giulia Di Benedetto, Claudia Carbone, Angela Bonaccorso, Giovanni Amato, Maria Josè Lo Faro, Chiara Burgaletto, Giovanni Puglisi, Renato Bernardini, Giuseppina Cantarella

**Affiliations:** 1Laboratory of Drug Delivery Technology, Department of Drug and Health Sciences, University of Catania, 95125 Catania, Italy; tmusumec@unict.it (T.M.); ccarbone@unict.it (C.C.); abonaccorso@unict.it (A.B.); gamato@unict.it (G.A.); gpuglisi@unict.it (G.P.); 2Department of Biomedical and Biotechnological Sciences, Section of Pharmacology, University of Catania, 95123 Catania, Italy; giulia.dibenedetto@unict.it (G.D.B.); chiaraburg@hotmail.it (C.B.); gcantare@unict.it (G.C.); 3Dipartimento di Fisica e Astronomia “Ettore Majorana”, Università di Catania, Via Santa Sofia 64, 95123 Catania, Italy; mariajose.lofaro@unict.it; 4CNR-IMM UoS Catania, Istituto per La Microelettronica e Microsistemi, Via Santa Sofia 64, 95123 Catania, Italy

**Keywords:** nanomedicine, polymers, lipids, monoclonal antibody, mice, nose-to-brain delivery, nanoparticles

## Abstract

Alzheimer’s disease (AD) is a neurodegenerative disorder that progressively compromises cognitive functions. Tumor necrosis factor (TNF)-Related Apoptosis Inducing Ligand (TRAIL), a proinflammatory cytokine belonging to the TNF superfamily, appears to be a key player in the inflammatory/immune orchestra of the AD brain. Despite the ability of an anti-TRAIL monoclonal antibody to reach the brain producing beneficial effects in AD mice, we attempted to develop such a TRAIL-neutralizing monoclonal antibody adsorbed on lipid and polymeric nanocarriers, for intranasal administration, in a valid approach to overcome issues related to both high dose and drug transport across the blood–brain barrier. The two types of nanomedicines produced showed physico-chemical characteristics appropriate for intranasal administration. As confirmed by enzyme-linked immunosorbent assay (ELISA), both nanomedicines were able to form a complex with the antibody with an encapsulation efficiency of ≈99%. After testing in vitro the immunoneutralizing properties of the nanomedicines, the latter were intranasally administered in AD mice. The antibody–nanocarrier complexes were detectable in the brain in substantial amounts at concentrations significantly higher compared to the free form of the anti-TRAIL antibody. These data support the use of nanomedicine as an optimal method for the delivery of the TRAIL neutralizing antibody to the brain through the nose-to-brain route, aiming to improve the biological attributes of anti-TRAIL-based therapy for AD treatment.

## 1. Introduction

Alzheimer’s disease (AD) is a progressive neurodegenerative disorder characterized by cognitive decline and the presence of two core pathologies, extracellular deposits of amyloid β (Aβ)plaques and intracellular neurofibrillary tangles (NFTs) [1,2,3].

Neuroinflammation represents one of the earliest pathomechanistic alterations throughout the AD continuum, where the immune-related genes and cytokines appear to be key participants [4,5].

Cytokines represent the major payload delivered by the main central and peripheral cell mediators of the inflammatory response in AD [6,7]. In particular, cytokines belonging to the TNF superfamily represent master regulators of the accelerated cell death rate which characterizes neurodegenerative processes, and play a crucial role in the initiation and orchestration of immunity and inflammation [8,9]. Among these, TRAIL stands out due to its prominent pro-apoptotic and immune-modulating properties in the complex scenario of AD etiopathogenesis [10,11,12].

While not constitutively expressed in the healthy human brain, sustained TRAIL immunoreactivity has been detected nearby Aβ plaques in human post-mortem AD brain [11], suggesting the importance to exploit TRAIL as a therapeutic target.

Consistently with these observations, the immunoneutralization of TRAIL demonstrated its efficacy by determining a rescue from the death of neurons challenged with Aβ in vitro [11]. Moreover, a significant functional recovery, decreased Aβ burden, as well as a rebalance of immune/inflammatory response overshoot and reduced tissue damage, have been shown in a transgenic mouse model of AD subjected to anti-TRAIL treatment [10,12].

Although previous data demonstrated the ability of the anti-TRAIL antibody to cross the blood–brain barrier (BBB) and reach the brain of the 3xTg-AD mice after its intraperitoneal injection [11], the adoption of innovative systems and a direct route for brain drug delivery could be a valuable strategy to increase its therapeutic efficacy.

Current available Food and Drug Administration (FDA)-approved pharmacotherapies for AD provide only symptomatic relief without any disease-modifying potential [13,14]. Licensed medicines are mostly administered via the oral and transdermal routes, entering the systemic circulation, and undergoing metabolic degradation which decreases their bioavailability [15,16,17]. In addition, in conventional formulation, drugs must cross several biological barriers, such as the BBB, before they get into the brain circulation [18,19].

Over the years, the intranasal (IN) route has been explored as an innovative, non-invasive, and alternative strategy for direct drug delivery to the brain via the nasal cavity [16,20,21]. The latter avoids the first-pass and peripheral side effects of the pharmacotherapy, improving its therapeutic efficacy [22,23]. On the other hand, lipidic and polymeric nanoparticles (NPs), have been envisioned as promising approaches to overcome some limits related to the drug administration into the nasal cavity, such as local enzymatic degradation and mucociliary clearance [24]. Indeed, nanomedicine provides a cutting-edge alternative approach to deliver a wide range of biological molecules (i.e., small molecules, proteins, peptides and gene materials) reducing the dose needed to obtain therapeutic effects [25].

The physico-chemical versatility, surface charge, shape, as well as the appropriate size of nanocarriers, are known to affect their ability to overcome biological membranes and the in vivo fate after administration [26,27,28,29].

Among several strategies tested to improve nasal adsorption, one of them was the use of lipid nanocarriers [30,31]. Nanostructured lipid carriers (NLC) with their lipophilic nature and small particle size are suitable for the nose-to-brain (N2B) delivery. Moreover, the encapsulation of the drug in the lipid matrix protects the molecule against enzymatic degradation [18,24,32].

Another feasible approach for drug delivery is represented by polymeric nanoparticles obtained by poly(lactic-co-glycolic acid) (PLGA), a biocompatible and biodegradable polymer, extensively used for the therapeutic delivery of proteins and peptides. Since decades, PLGA has been used for controlled drug release purposes and represents, therefore, an obvious choice for encapsulation of several molecules used in brain disorders [33].

Considering that nanomedicine and IN route represent interesting approaches for brain targeting [21], we aimed to develop and compare lipidic and polymeric nanomedicines, for more effective delivery of a TRAIL-neutralizing monoclonal antibody to the brain via the N2B route and successful treatment of neurodegenerative disorders, such as AD.

## 2. Materials and Methods

### 2.1. Materials

Cetyl palmitate (Cutina CP) was a gift from BASF Italia S.p.A. (Cesano Maderno, Monza e Brianza, Italy). Tegin O (Gliceryl Monooleate), Brij 98 (Oleth-20) and Isopropyl stearate (IPS) were purchased from A.C.E.F. S.p.a. (Piacenza, Italy). Didodecyldimethylam-monium bromide (DDAB) was bought from Sigma–Aldrich (Milan, Italy). PLGA was purchased from Boehringer Ingelheim (Ingelheim am Rhein, Germany). The TRAIL-neutralizing antibody (purified rat anti-mouse CD253) was obtained from BD Biosciences (San Jose, CA, USA). Ultrapure water was used throughout this study. Tween 80^®^ was purchased from Sigma-Aldrich (Milan, Italy). All other chemicals used for the study were of analytical grade, and were purchased from Analyticals, Carlo Erba. Cell culture media were obtained from Life Technologies Corporation (Grand Island, NY, USA). All other compounds were of the highest commercial grade available.

### 2.2. Preparation of Nanoparticles

Polymeric nanoparticles (PLGA NPs or NANO-A) were prepared by solvent displacement followed by polymer deposition, as previously reported. Briefly, the polymer (30 mg) was dissolved in the organic solvent (acetone). The aqueous phase was composed by H_2_O/EtOH (1:1 *v*/*v*) added with Tween 80^®^ as surfactant at the concentration of 0.1% *w*/*v*. At 25 °C the organic phase was dropped in the aqueous one under magnetic stirring at ratio 1:2 (acetone: H2O/EtOH *v*/*v*). At the end a milky colloidal suspension was obtained. The organic phase was removed by rotavapor (Buchi, Cornaredo, Italy) at 40 °C. At this step the purification process was carried out by Thermo Scientific SL 16 R Centrifuge (Thermo Scientific Inc., Waltham, MA, USA) at 15,777× *g* for 1 h at 8 °C. Pellet was collected and resuspended in water containing the 5% *w*/v of glucose as cryoprotectant and then were frozen and freeze-dried for 24 h (Freeze Dryer Edwards Modulyo, Akribis Scientific Limited, Knutsford, Cheshire, UK).

A low-energy organic solvent-free phase inversion process (PIT method) was used for the preparation of the lipid nanoparticles (NLC NPs or NANO-B). A total amount of 13.1% *w*/*w* of the surfactant mixture tegin O/brij 98 was used in combination with 5% *w*/*w* of solid and liquid lipid (4:1). 0.5% *w*/*w* of the cationic lipid DDAB was used to obtain positively charged lipid nanoparticles. In order to remove the excess of surfactants, NLC NPs were centrifuged using a Thermo Scientific SL 16 R Centrifuge (Thermo Scientific Inc., Waltham, MA, USA) at 2168× *g* for 1 h at 10 °C.

The freeze-dried NANO-A and pellet of NANO-B NPs were re-suspended in 1 mL of physiological solution and incubated at 4 °C for 24 h with 100 µL of the anti-TRAIL monoclonal antibody (50 µg/mL) to form the NANO-A complex and NANO-B complex. After the incubation time, NANO-A complex and NANO-B complex were purified to remove the not adsorbed antibody by ultracentrifugation (15000× *g*) for 15 min at 10 °C, through Thermo Scientific SL 16 R Centrifuge (Thermo Scientific Scientific Inc., Waltham, MA, USA).

### 2.3. Characterization of Nanoparticles

#### 2.3.1. Mean Particle Size, PDI and Zeta Potential Determination

The mean particle size (Z-Ave), the polydispersity index (PDI), and the Zeta-potential (ZP) of the NPs were measured at 25 °C by photon correlation spectroscopy (PCS) using a Zetasizer Nano ZS (Malvern, Malvern, UK). Each sample was analyzed after dilution (50 µL in 1 mL of ultrapure water) into disposable sizing cuvettes (DTS 0012). Results are shown as mean of at least three measurements ± standard deviation (SD).

The osmolarity of the NANO-A and NANO-B complexes were determined through a digital osmometer (Osmomat 030, Gonotec, Berlin, Germany) after its calibration with distilled water and sodium chloride (0.9% *w*/*v*). The determination of pH was carried out using a pH-meter at 25 °C (Checket, Hanna Instrument, Woonsocket, RI, USA) which was calibrated before each use with 3 buffer solutions. For both measurements the obtained values for each sample are the mean of 3 different measurements.

#### 2.3.2. Scanning Electron Microscopy (SEM)

The surface morphology of NPs synthesized was assessed using scanning electron microscopy (SEM). Both lipidic and polymeric nanoparticles were prepared for the electron microscope observation with a spin-coating procedure at 500 rpm for 1 min with a Suss Microtech instrument and left to air dry for a few hours. To ensure good conductivity, prior to the analysis, all the SEM samples were subjected to gold sputtering at a pressure of 10^−3^ mbar with an Emitech K500X equipment, for an approximately 5 nm coating. The SEM images were acquired at a low voltage of 3 KV with an InLens detector by using a Field Gemini microscope from Zeiss.

#### 2.3.3. Encapsulation Efficiency through Enzyme-Linked Immunosorbent Assay (ELISA)

The encapsulation efficiency (EE%) was determined as reported in the Equation (1). In particular, it quantifies the amount of not adsorbed antibody in the surnatant recovered after centrifugation process measured by ELISA assay, and the adsorbed amount was determined as difference between the initially added amount and the not adsorbed fraction.

The wells of microtiter plates were coated with recombinant mouse TRAIL protein (Enzo Life Sciences, Inc., Farmingdale, NY, USA) diluted in 0.1 mM carbonate–bicarbonate coating buffer (pH 9.6). The plates were blocked with 5% BSA for 1 h at room temperature (RT) and then incubated with the free anti-TRAIL, anti-TRAIL adsorbed on NPs, and empty NPs as a control in phosphate-buffered saline (PBS) containing 1% BSA for 1 h at RT. Wells were washed with PBS with Tween 20 (PBS-T), incubated with the HRP-conjugated anti-mouse IgG (1:20.000) for 1 h, then washed with PBS-T, and finally incubated with the horseradish peroxidase substrate (TMB solution) for 20 min at 37 °C. The reaction was stopped by HCl 2N. Absorbance was read at a wavelength of 450 nm by using a microplate reader (Bio-Rad, Hercules, CA, USA). Each measurement was carried out at least in triplicate. The EE% was calculated by the following Equation (1) [34]:
(1)
$$EE\%=\frac{Total\;amount\;of\;antibody\;added-Amount\;of\;not\;adsorbed\;antibody}{Total\;amount\;of\;antibody\;added}*100$$


### 2.4. In Vitro Evaluation of Cytotoxicity of Nanoparticles

#### 2.4.1. Cell Culture

The murine macrophage RAW 264.7 cell line was purchased from American Type Culture Collection (ATCC, TIB-71; Manassas, VA, USA) and maintained in Dulbecco’s Modified Eagle Medium (DMEM) supplemented with 10% heat-inactivated fetal bovine serum (FBS), 100 μg/L streptomycin, and 100 IU/mL penicillin at 37 °C in a 5% CO_2_/95 % atmosphere.

#### 2.4.2. Cell Viability Assay

Cell viability was determined by using 3-[4,5 dimethylthiazol-2-yl]-2,5-diphenyltetrazolium bromide (MTT) assay (Sigma-Aldrich, Milan, Italy). Briefly, RAW 264.7 cells were seeded into 96-well plates at a density of 5 × 10^5^ cells/mL. After 72 h of treatment, cell viability was measured by the reduction of MTT solution (0.5 mg/mL). After 3 h of incubation at 37 °C, the MTT solution was removed, and dimethyl sulfoxide (DMSO) was added to obtain cell lysis and solubilization of blue formazan crystals resulting from MTT reduction by viable cells’ mitochondrial activity. The optical density of the formed blue formazan was measured at 545 nm using a microplate reader (Bio-Rad, Hercules, CA, USA). Data were expressed as the mean percentage of viable cells versus control.

### 2.5. In Vivo Studies

#### 2.5.1. Animals

Male 3xTg-AD [B6129-Psen1tm1MpmTg (APPSwe, tauP30L) 1Lfa/J] [35] and wild-type mice (B6129SF2/J) were purchased from The Jackson Laboratory (Bar Harbor, ME, USA). The 3xTg-AD, overexpressing mutant amyloid precursor protein (APP (APPSwe)), presenilin 1 (PSEN1 (PS1M146V)), and hyperphosphorylated tau (tauP301L), were originally generated by co-injecting two independent transgene constructs encoding human APPSwe and tauP301L (4R/0 N) (controlled by murine Thy1.2 regulatory elements) into single-cell embryos harvested from mutant homozygous PS1M146V knock-in mice. Wild-type mice of mixed genetic background 129/C57BL6 were used as controls. These mice have been characterized and described by Oddo et al. [35]. The animals were maintained on a 12-h light/dark cycle in a temperature- and humidity-controlled rooms and food and water were available ad libitum.

#### 2.5.2. Experimental Groups and Intranasal Drug Administration

For intranasal drug administration study, twenty-five 3xTg-AD and twenty-five wild-type mice were enrolled at 12 months of age and the following groups were established: (1) 3xTg-AD or wild-type (WT) mice plus TRAIL-neutralizing antibody (Purified Rat Anti-Mouse CD253) (concentration: 0.05 mg/mL; 200 μL/mouse); (2) 3xTg-AD or WT mice plus NANO-A; (3) 3xTg-AD or WT mice plus NANO-B (the unloaded nanosystems were tested at the same concentrations that correspond to NANO-A complex and NANO-B complex); (4) 3xTg-AD or wild-type mice plus NANO-A complex; and (5) 3xTg-AD or WT mice plus NANO-B complex (50 µg/mL of TRAIL-neutralizing antibody). Animals (*n* = 5 per experimental group) were sacrificed after 24 h.

### 2.6. Immunofluorescence

To detect brain localization of the TRAIL-neutralizing monoclonal antibody (free or adsorbed onto NANO-A or NANO-B NPs) intranasally administered, brains were collected and fixed overnight in 10% neutral buffered formalin (Bio-Optica, Milan, Italy). After overnight washing, the samples were dehydrated in graded ethanol and paraffin-embedded taking care to preserve their anatomical orientation. Sections were then cut in the coronal plane and 5- µm thick sections were then obtained by routine procedures, mounted on silanized glass slides, and air-dried. Immunofluorescence was performed using a goat anti-rat IgG fluorescein-conjugated antibody (1:200; Merck Millipore, Burlington, MA, USA) at dark for 1 h at RT. Finally, for nuclear staining and stabilization of fluorescent signals, slides were washed and mounted with DAPI-containing mounting solution (Fluoroshield with DAPI; Sigma-Aldrich, Milan, Italy) and secured with a coverslip. All images were observed using an epifluorescent Zeiss Observer.Z1 microscope (Zeiss, Oberkochen, Germany). Densitometric count of fluorescent signal was performed using an ImageJ software (Available online: https://imagej.nih.gov/ij/ (accessed on 12 January 2022)) and represented as integrated density (% of control).

### 2.7. Statistical Analysis of Results

Data were analyzed by the one-way analysis of variance (ANOVA) test, followed by the Fisher’s Least Significant Difference test. Results were reported as mean ± SD. Differences between groups were considered significant for *p*-value < 0.05 or < 0.01. Analyses were performed using Prism 8 (GraphPad software, Inc., La Jolla, CA, USA).

## 3. Results

### 3.1. NLC and PLGA Polymeric Nanocarriers: Physico-Chemical and Morphological Characterization of Nanosystems

Considering that several parameters, such as physico-chemical versatility, mean size, surface charge and shape of the NPs could influence cellular uptake [36,37], different types of nanosystem should be investigated to obtain an efficient drug delivery via the N2B route. Consistent with the latter and with the effect of protein adsorption onto the surface of NPs, we aimed to develop and compare two different formulations: polymeric or PLGA NPs (NANO-A) and lipidic (nanostructured lipid carriers or NLC, NANO-B) nanomedicines. Both lipidic and polymeric nanosystems were widely investigated for a more efficient brain delivery of active drugs through IN route [21,24]. To preserve the physico-chemical properties of the monoclonal antibody, the selected preparation methods for both nanomedicines were slightly modified compared to the known procedure which allowed to load small molecules. The two studied nanomedicines adsorbed onto their surface the anti-TRAIL monoclonal antibody, adding it at the end of the preparation procedure. The freeze-drying process was applied to convert samples from aqueous suspensions to dried powders, and cryoprotectant agent was used before the lyophilization procedure to preserve the physico-chemical properties of the colloidal systems. The cryoprotectant used was glucose, according to Musumeci et al. that demonstrated its efficacy in PLGA nanoparticles [38].

SEM micrographies of empty and anti-TRAIL adsorbed on NPs surface showed well-isolated, non-adhered, and round-shaped NPs with smooth surfaces and relatively regular sizes (Figure 1). Results also demonstrated that the presence of the antibody did not influence the shape of nanomedicine.

Physico-chemical characterization has shown significant differences between the two types of nanocarriers studied. The mean particle size (Z-Ave), the polydispersity index (PDI), and the Zeta-potential (ZP) of the NPs are provided in Table 1.

Indeed, photon correlation spectroscopy (PCS) analysis demonstrated that NLC or NANO-B NPs increased their mean particle size from ~30 nm of the unloaded to ~250 nm of the NANO-B complex (Figure 1a,b; Table 1). This phenomenon is due to the adsorption of the macromolecules on the particle surface. Moreover, the PDI values (<0.3) indicated a homogeneous particle size distribution of the NPs.

Regarding PLGA, unloaded nanocarriers (NANO-A) showed a particle mean size of 453.50 ± 8.42, which was however slightly increased (543.20 ± 15.78 nm) when the anti-TRAIL monoclonal antibody was adsorbed onto the particle surface to form the NANO-A complex (Figure 1c,d). NANO-A complex showed a slight degree of non-uniformity of a size distribution of particles, as evidenced by PDI values of about 0.4.

Both the NPs increased their mean size, but this variation is more evident for NANO-B complex compared to NANO-A complex as shown in Figure 1.

Generally, the addition of a coating layer with the positively charged cationic lipid such as Didodecyldimethylammonium bromide (DDAB) is used to enhance mucoadhesive properties. In this case the positive shell influences the deposition of the monoclonal antibody on the NANO-B particle surface.

The effective adsorption of the anti-TRAIL monoclonal antibody on the NPs surface was evidenced by the decrease in absolute values of ZP for both NANO-A and NANO-B complexes with respect to the unloaded ones, as demonstrated by other Authors [39,40]. In addition, although a variation occurred also in the surface charge of NANO-A system (from −29 mV to −19 mV), the ZP falled into the range of negative values.

The entrapment efficiency of the anti-TRAIL monoclonal antibody (50 µg/mL) in the NPs prepared was calculated for both the nanosystems investigated. The data collected are shown in Table 1.

Results of these measurements obtained by means of an ELISA, confirmed that both nanomedicines were able to form a complex with the antibody with an encapsulation efficiency of ≈99% (Table 1).

Such a high value of EE% obtained could be related to the complete adsorption of the antibody onto the surface of the particles and demonstrate the potential usefulness of the nanocomplexes for the delivery of the macromolecule through the N2B route.

### 3.2. Cell Viability of RAW 264.7 Cells Treated with TRAIL

Subsequently, with the aim to assess a suitable in vitro model to test the effect of nanosystems developed, we first tested TRAIL toxicity on the murine macrophage RAW 264.7 cell line, to see whether the TRAIL neutralizing monoclonal antibody was able to prevent TRAIL effects.

With this purpose, cell cultures were challenged with TRAIL (100 ng/mL) for 72 h.

As shown in Figure 2, TRAIL significantly reduced the viability of RAW 264.7 cells compared to the untreated cells, consistently with previous observations [41]. Preincubation of cells with TRAIL-neutralizing monoclonal antibody at the concentration of 1 µg/mL in the presence of TRAIL (100 ng/mL) resulted in a total rescue of the cells from the proapoptotic effect of the cytokine, as previously demonstrated also in another in vitro model [11].

### 3.3. Efficacy of NANO-A or NANO-B Complexes in Preventing TRAIL toxicity in RAW 264.7 Cells

The NPs system must be able to deliver active drugs to the target site without compromising cell viability. In light of the ability of anti-TRAIL to rescue RAW 264.7 cells from the cytotoxic effect induced by TRAIL, we tested the tolerability of both the negatively charged NANO-A and positively charged NANO-B NPs in the same cell line. Unloaded NANO-A and NANO-B did not show toxicity at the concentration used (50 µg/mL) and were well-tolerated, demonstrating the suitability of these systems as anti-TRAIL nanocarriers (Figure 3). Subsequently, we investigated the efficacy of anti-TRAIL complexed with either nanoparticle NANO-A or NANO-B. RAW 264.7 cells were incubated for 72 h with TRAIL (100 ng/mL), alone or in combination with both anti-TRAIL complexed with NANO-A or NANO-B (NANO-A or NANO-B complexes). Results demonstrated that both NANO-A and NANO-B complexes were able to prevent the TRAIL-induced cell death in RAW 264.7 cells, thus confirming the efficacy of both antibody complexes (Figure 3).

### 3.4. Immunofluorescence Detection of Brain Levels of NANO-A or NANO-B Complexes

To investigate whether NANO-A or NANO-B complexes were more effective in brain targeting compared to the free anti-TRAIL antibody, 12-month-old 3xTg-AD and age-matched wild-type mice were intranasally administered with anti-TRAIL antibody free (0.05 mg/mL; 200 μL/mouse) or conjugated with NANO-A or NANO-B NPs (50 µg/mL). Animals were sacrificed 24 h after treatment and brains were promptly removed for immunofluorescence analysis.

Then, to verify whether after intranasal administration the NANO-A and NANO-B complexes effectively reached the brain in substantial amounts, immunofluorescence was performed on histological specimens from hippocampi (Figure 4).

Immunofluorescence images revealed the presence of low levels of the naïve antibody in the brain of both WT and 3xTg-AD mice, whereas the hippocampus of mice treated with NANO-A or NANO-B complex showed a significantly increased level of both complexes (Figure 5).

## 4. Discussion

In the present study, we investigated the ability of a TRAIL neutralizing monoclonal antibody adsorbed onto the surface of PLGA and NLC NPs to reach the brain through the nose-to-brain (N2B) route, to obtain more effective drug delivery systems suitable for the treatment of neurodegenerative disorders, such as AD.

Currently, intranasal administration represents a widely investigated strategy to bypass the BBB, which is the most important limiting factor for the efficient delivery of active drugs to the brain [42,43,44]. This innovative strategy was first brought about in 2008 by Frey and collaborators [45,46], followed by a worldwide growing interest also in the European Union, which has heavily funded a number of projects, such as, for instance, the N2B-patch project in 2017 [47].

Nanomedicine constitutes a valid approach to overcome some limits related to drug administration into the nasal cavity, such as local enzymatic degradation and mucociliary clearance [48,49], and to deliver biological molecules (i.e., peptides or proteins), reducing the dose needed to obtain therapeutic effects [50].

In previous works, we demonstrated that TRAIL represents an important target in the AD brain [10]. In fact, TRAIL immunoreactivity has been detected nearby Aβ plaques in post-mortem human AD brains [11]. Consistently, neutralization of TRAIL is associated with rescue from the death of human neurons challenged in vitro with Aβ, as well as with reduced Aβ burden, significant functional recovery, and a restrain of immune/inflammatory response overshoot and consequent tissue damage in a triple transgenic mouse model of AD [10,11,12].

We further demonstrated that the anti-TRAIL antibody reaches the brain when injected intraperitoneally in 3xTg-AD mice, confirming that IgG2 are able to cross the mouse BBB [11,51].

Now, in the present study, we propose that the combination of nanocarriers and N2B route [21] could offer an effective advantage to optimize the efficacy of drugs, both small molecules and macromolecules, such as antibodies, that are then allowed to attain the brain in substantial amounts.

Over the years, different types of nanosystems have been investigated for this purpose and several factors, such as the type of nanocarriers, mean particle size and surface charge, can influence cellular uptake of the nanocarriers and their distribution after in vivo administration [52]. Indeed, a number of pathways are involved in the N2B delivery after the deposition of nanomedicine in the nasal cavity (trigeminal, olfactory and systemic), affecting also their distribution in several cerebral regions [53].

In this scenario, we propose to develop and compare two nanosystems for the delivery of the anti-TRAIL antibody into the brain.

The use of lipid nanoparticles, such as nanostructured lipid carriers (NLC), in nasal formulations, have shown promising outcomes on a wide array of indications such as brain diseases, including, gliomas, epilepsy, Parkinson’s disease, and AD [54].

Moreover, Musumeci and colleagues showed that a coated shell of chitosan onto the PLGA nanoparticles confers to the NPs mucoadhesive properties and positive charge and can overcome the limitations due to the administration site. In fact, this strategy could augment adsorbtion of molecules across the mucosal barriers [55] by increasing their residence time in the nasal cavity [56], but at the same time it could delay the entry of nanosystems into the brain [53].

In this line, several factors in the NPs formulation and freeze-drying process need to be optimized to obtain a suitable product [57]. It is known that the addition of a cryoprotective agent to the empty nanocarrier before the lyophilization procedure represents a suitable method to preserve the final physico-chemical properties of the sample. According to our previous work which showed the efficacy of different sugars on preservation properties of PLA and PLGA nanoparticles [38], in the present study, we selected glucose as a cryoprotectant for its high biocompatibility, low cost of raw material, and the absence of other biases, such as potential penetration enhancer effect correlated with the use of cyclodextrins [58].

SEM scans of empty and anti-TRAIL-loaded NPs showed well-separated NPs with a spherical shape, smooth surface, and regular sizes. In addition, the adsorbtion of the antibody did not influence the shape of the nanomedicine, thus confirming the suitability of these nanosystems for the delivery of anti-TRAIL into the brain.

Both NPs showed increased mean size, but such variation was more evident for the NANO-B complex compared to the NANO-A complex. The presence of DDAB in the NANO-B determined the positive charge for the system that influence the deposition of the monoclonal antibody on the particle surface. The effective adsorbtion of the monoclonal antibody is demonstrated by the reduction in absolute values of ZP for both NANO-A and NANO-B complexes, compared to the unloaded ones, as already demonstrated by different authors [39,40]. We could hypothesize that in the case of NANO-A complex, the antibody interacts with the fraction of amino-acids that showed positive charged residues, in opposite in the NANO-B complex the negatively charged amino-acids fraction of the protein interacts with the surface of NPs.

In in vitro experiments, we confirmed the capability of the TRAIL-neutralizing antibody to block the toxic effect of TRAIL in cell cultures. Consistent with previous data which demonstrate the toxicity of TRAIL on human neural cells in vitro [10], MTT assay showed that TRAIL is also able to kill RAW 264.7 cells and that such effect is prevented by the anti-TRAIL monoclonal antibody.

Indeed, as reported by Warat and colleagues, TRAIL is able to kill RAW264.7 cells through its death receptor DR5 [59]. Moreover, negatively charged empty NANO-A and positively charged empty NANO-B, both suitable for N2B delivery, did not affect RAW264.7 cell viability, suggesting an adequate tolerability for both nanomedicines.

Similarly to the free TRAIL antibody, the NANO-A complex, as well as the NANO-B complex were both able to efficiently prevent the TRAIL-induced cell death in RAW cells. Interestingly, significant TRAIL-neutralizing efficacy was observed for both complexes in RAW264.7 cell cultures.

To understand the real biological value of our data, we performed an in vivo treatment with NANO-A and NANO-B complexes. Excitingly, both NANO complexes, with a better performance by NANO-B, were able to reach the brain and were abundantly localizing in the CA2 and CA3 hippocampal areas of 3xTg-AD mice.

Scanty data are available about the ability of antibody-bearing nanoparticles to cross the BBB when administered intranasally. Indeed, Chu et al. [60] demonstrated that anti-EPHA3-modified NPs may potentially serve as a N2B drug carrier for the treatment of glioblastoma, confirming the concept that these systems may well attain the brain tissue to deliver the drug.

Here, we show that a fair amount of specific immunofluorescence is diffused in the CNS of 3xTg-AD mice after intranasal administration, suggesting that both NANO-A and NANO-B complexes may efficiently achieve their target, TRAIL, richly represented in the inflammatory foci in the hippocampus of these animals [61].

As previously demonstrated mean size can affect the direct transport through the olfactory or trigeminal axonal nerves. The difference in the mean size evidenced between the two nanocomplexes deeply influenced their distribution in hippocampus after intranasal administration. Indeed, the nanosystems with the lowest mean size are more evident in the cerebral investigated area.

These data also supported the hypothesis that the mean size influences the distribution in the brain more compared to the zeta potential values. In addition, the positive charge of NLC did not affect their localization into the brain, and no retardation effect was shown.

Here, for the first time in a mouse model of AD, we report that it is possible to successfully deliver a monoclonal antibody known to block one among major inflammatory factors in the AD brain, TRAIL. In the present work, the developed nanocomplexes efficiently reached the brain, where it has already been demonstrated that the blockade of TRAIL brings about dramatic cognitive recovery in 3xTg-AD mice, stopping the progression of symptoms, as well as that of brain pathology [10,11].

Finally, delivering an anti-TRAIL antibody adsorbed onto the surface of PLGA and NLC NPs that rapidly and efficiently reaches the brain after intranasal administration, may represent an advantageous and easy tool for the non-invasive and effective treatment of AD.

## Figures and Tables

**Figure 1 biomedicines-10-00985-f001:**
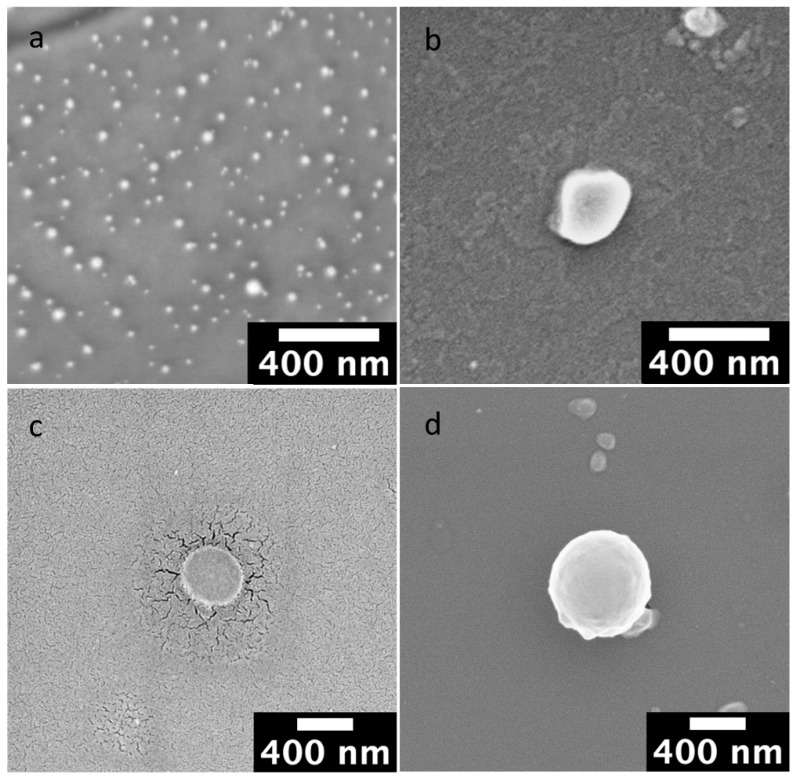
Scanning electron microscopy (SEM) micrographies of (**a**) NANO-B (Scale bar = 400 nm); (**b**) NANO-B complex (Scale bar = 400 nm); (**c**) NANO-A (Scale bar = 400 nm); (**d**) NANO-A complex (Scale bar = 400 nm).

**Figure 2 biomedicines-10-00985-f002:**
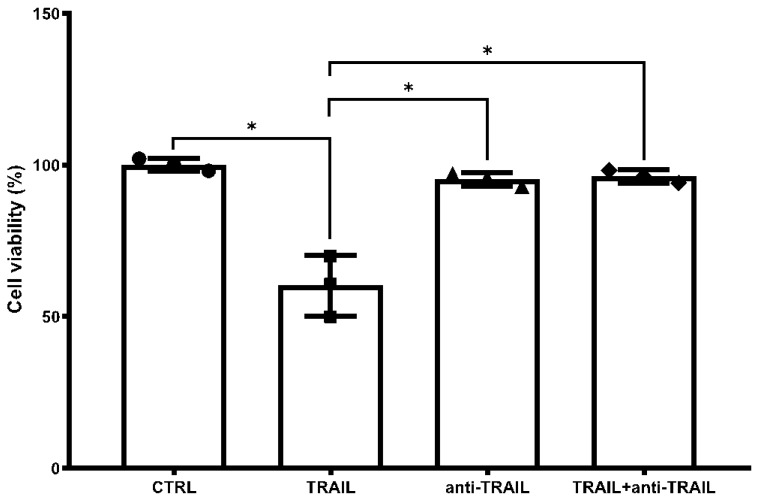
Cell viability percentage after 72 h of treatment with TRAIL (100 ng/mL), anti-TRAIL (1 μg/mL) or the combination of the compounds. Data are expressed as mean ± SD. One-way ANOVA followed by the Fisher’s LSD test were used for statistical analysis. * *p* < 0.05.

**Figure 3 biomedicines-10-00985-f003:**
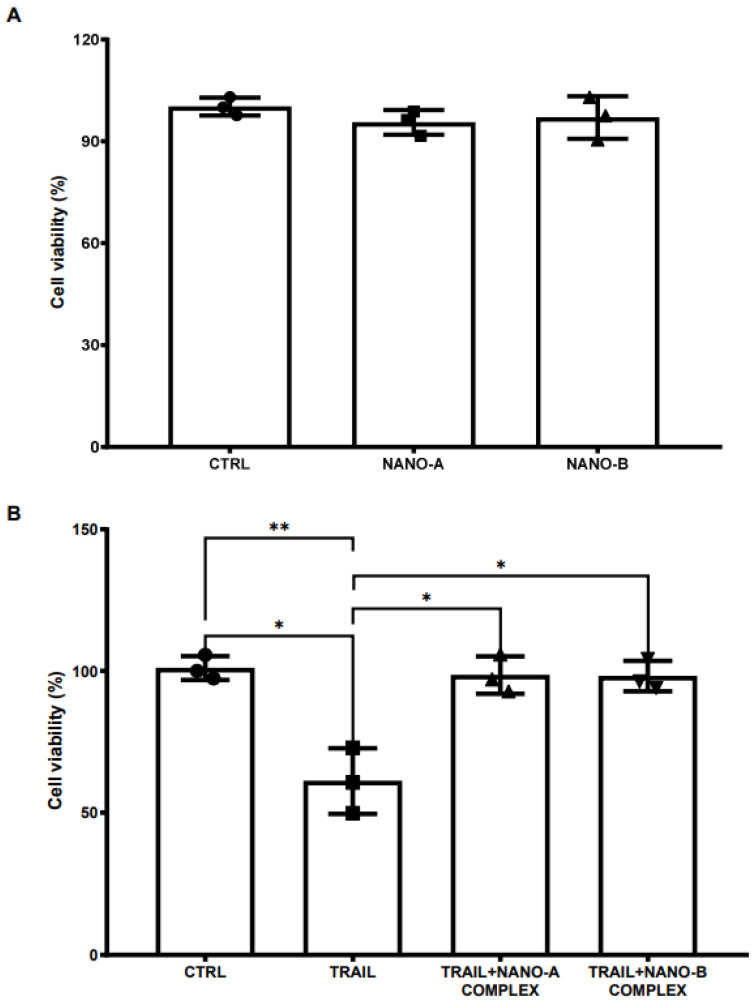
RAW 264.7 cell viability (%) after 72 h of treatment with empty nanoparticles (NANO-A and NANO-B) (**A**). In (**B**) is depicted the cell viability percentage after treatment with TRAIL (100 ng/mL) alone or in combination with either nanoparticles complexed with the anti-TRAIL antibody (NANO-A or NANO-B complexes). Data are expressed as mean ± SD. One-way ANOVA followed by the Fisher’s LSD test were used for statistical analysis. * *p* < 0.05 or ** *p* < 0.01.

**Figure 4 biomedicines-10-00985-f004:**
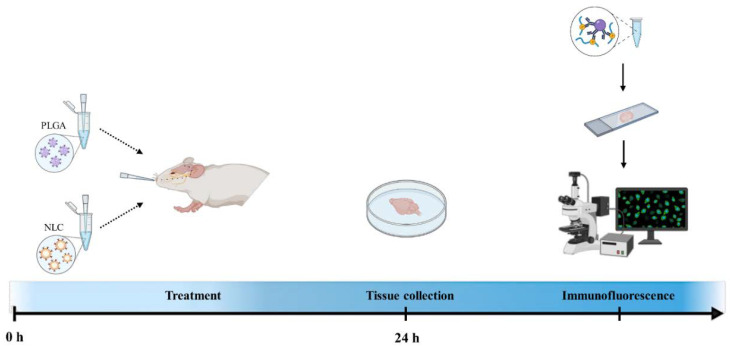
Experimental design for the nanoparticles administration. Twelve-month-old wild-type (WT) and 3xTg-AD mice were intranasally administered with both empty and anti-TRAIL loaded nanoparticles (PLGA or NANO-A and NLC or NANO-B NPs). After 24 h, brains were collected for immunofluorescence analysis. Created with www.BioRender.com (accessed on 10 February 2022).

**Figure 5 biomedicines-10-00985-f005:**
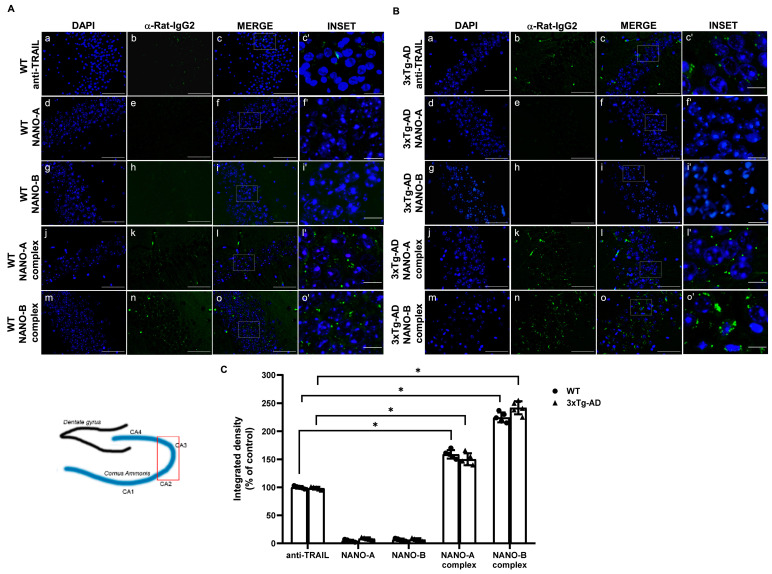
Comparison of polymeric or lipidic nanoparticles complexed with anti-TRAIL in the hippocampus of 3xTg-AD mice after intranasal administration. Representative immunofluorescence microscopy images showing the localization of the anti-TRAIL monoclonal antibody in the hippocampal sections from WT (**A**) or 3×Tg-AD (**B**) mice intranasally administered for 24 h with the anti-TRAIL monoclonal antibody, empty, or anti-TRAIL-loaded NANO-A, NANO-B nanoparticles. The fluorescence signal was detected by using a fluorescent secondary anti-rat IgG. The insets represent the respective areas magnified. (**C**) The densitometric count of fluorescence was performed with the aid of ImageJ software (available online: https://imagej.nih.gov/ij/ (accessed on 12 January 2022)) and represented as integrated density (% of control). Data are expressed as the mean ± SD. One-way ANOVA followed by Fisher’s LSD test was used for statistical analysis. * *p* < 0.05. (a–o scale bar = 50 µm; c’, f’, i’, l’, o’ scale bar = 10 µm).

**Table 1 biomedicines-10-00985-t001:** Physico-chemical and technological characterization of nanomedicines and nanocomplexes obtained with polymeric (NANO-A) and lipidic (NANO-B) raw materials. Each value is expressed as mean ± SD from at least three measurements.

SAMPLES	Z-Ave (nm±SD)	PDI ± SD	ZP (mV ± SD)	% EE
NANO-A	453.50 ± 8.42	0.385± 0.01	−29.0 ± 0.05	-
NANO-A complex	543.20 ± 15.78	0.304 ± 0.05	−19.1 ± 0.49	99.81 ± 0.03
NANO-B	31.40 ± 3.72	0.201 ± 0.02	54.50 ± 0.32	-
NANO-B complex	242.90 ± 44.96	0.208 ± 0.086	37.60 ± 0.45	99.71 ± 0.02

## Data Availability

The data presented in this study are available from the corresponding author on reasonable request.

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
