# Peer review of "Intranasal Administration of a TRAIL Neutralizing Monoclonal Antibody Adsorbed in PLGA Nanoparticles and NLC Nanosystems: An In Vivo Study on a Mouse Model of Alzheimer’s Disease"

_biomedicines, 2022, doi:10.3390/biomedicines10050985_

Round 1

Reviewer 1 Report

  1. Authors should describe the synthesis of NANO-A and NANO-B elaborately as it is unclear from the methods part.
  2. The encapsulation efficiency calculation method is unclear from the methods section. 
  3. Authors should elaborately describe the purification methods from the unentrapped antibodies, as both formulations are different. 
  4.  The authors used cryopreservation for storing the prepared nanoparticles. However, when the nanoparticles are reconstituted, the stability of adsorbed antibodies needs to be verified as the electrostatic attractions and weak attractive forces are usually unstable during cryopreservation. Authors need to perform stability studies before and after cryopreservation.
  5. Authors have used "absorbed" and "Adsorbed" interchangeably throughout the manuscript. Both processes are completely different. Authors should be consistent in using the type of antibody interaction with the nanoparticles and clearly demonstrate the rationale of such type of interactions.
  6. Authors should clearly show the forces behind the adsorption of antibodies to the nanoparticles. For instance, NANO-A is a negatively charged particle and NANO-B is positively charged. The mechanisms of adsorption in both the formulations are different, which was not showed.
  7. The rationale for selecting RAW 264.7 cell line for in vitro studies is unclear as it's not the target for the formulations.
  8. Authors need to provide a high-resolution immunofluorescence images with low and higher magnifications to clearly visualize the fluorescent signal. The authors should provide a scale bar for the fluorescence images.
  9. The authors should provide mechanistic insights into the nanoparticle formulations used for the study. Both formulations differ in their physicochemical characteristics. However, both of them have crossed the BBB and are effective via the intranasal route. 

Author Response

Reviewer(s)' Comments for the Author:

Reviewer #1 Comments and Suggestions for Authors:

Q1. Authors should describe the synthesis of NANO-A and NANO-B elaborately as it is unclear from the methods part.

Q3. Authors should elaborately describe the purification methods from the unentrapped antibodies, as both formulations are different. 

R.1, R.3) Thank to reviewer for these comments. As requested, a more detailed description regarding preparation method and purification processes has been included in the appropriate section of the manuscript. The modified text is now added in the proper section.

Q2. The encapsulation efficiency calculation method is unclear from the methods section. 

R.2) As suggested by the referee, we modify the text in this section. We hope that now the explanation of the calculation method is clearer.

Q4. The authors used cryopreservation for storing the prepared nanoparticles. However, when the nanoparticles are reconstituted, the stability of adsorbed antibodies needs to be verified as the electrostatic attractions and weak attractive forces are usually unstable during cryopreservation. Authors need to perform stability studies before and after cryopreservation.

R.4) We apologize for the imprecise description of this part. We modify the text to better describe the aim of our preparation method. We did not freeze-dry the nanoparticles for the long storage stability but to obtain a final product that have the ability to adsorb the antibody onto its surface. For this reason, we did not perform stability studies on antibody before and after cryopreservation. We also agree with Referee that it is possible that some steps in the preparation procedure could destabilize the antibody and for this reason we added it at the end of the procedure. Moreover, in this case, the freeze-drying process allowed us to obtain a concentrated product that has been administrated intranasally that requires a very little volume of instilled product.

Q5a. Authors have used "absorbed" and "Adsorbed" interchangeably throughout the manuscript. Both processes are completely different.

R.5a) We apologize with the Referee; we have amended the inappropriate words previously indicated throughout the manuscript. In our work antibody is adsorbed onto the nanoparticles surface. We thank the Referee for this suggestion because it has allowed us to also revise the term previously reported in the Title.

Q5b. Authors should be consistent in using the type of antibody interaction with the nanoparticles and clearly demonstrate the rationale of such type of interactions.

Q6. Authors should clearly show the forces behind the adsorption of antibodies to the nanoparticles. For instance, NANO-A is a negatively charged particle and NANO-B is positively charged. The mechanisms of adsorption in both the formulations are different, which was not showed.

R.5b, R.6) Referred to the type of the interactions, we agree with Referee that the nanosystems are different in terms of surface charge. Currently, we do not have sufficiently information about the amino acids sequence of this antibody that we have also requested to the BD Biosciences Company.

Without this information, we could speculate that in the case of NANO-A complex the antibody interacts with the fraction of amino-acids that showed positive charged residues and, on the contrary, in the NANO-B complex the negatively charged amino-acids fraction of the protein interacts with the surface of NPs. This concept was also reported in the proper section of the manuscript

Q7. The rationale for selecting RAW 264.7 cell line for in vitro studies is unclear as it's not the target for the formulations.

R.7) Thank you for your comment. The rationale for selecting RAW 264.7 cell line for in vitro studies is aimed at evaluating whether the TRAIL neutralizing antibody adsorbed onto the nanoparticle surface to form either NANO-A or NANO-B complexes is able to protect the cells from the toxicity induced by the cytokine TRAIL.

In addition, the fact that the main source of TRAIL basically comes from the central nervous system-infiltrated macrophages, and that the latter are sensitive to TRAIL-induced apoptosis due to the expression of the TRAIL death receptor (Huang et al., Cell Mol Immunol. 2005; Lee et al., Endocrinology. 2019), further support the rationale of using this in vitro model. 

Q8. Authors need to provide a high-resolution immunofluorescence images with low and higher magnifications to clearly visualize the fluorescent signal. The authors should provide a scale bar for the fluorescence images.

R.8) Following the Referee suggestions, we provide high-resolution immunofluorescence images with low and higher magnifications in the Figure 5. Now the scale bars for fluorescence images have been added in correspondent Figure.

Q9. The authors should provide mechanistic insights into the nanoparticle formulations used for the study. Both formulations differ in their physicochemical characteristics. However, both of them have crossed the BBB and are effective via the intranasal route. 

R.9) Thanks to the Referee for her/his comments. For the sake of clarity, and as already reported in the Manuscript, we attempted to develop a TRAIL-neutralizing monoclonal antibody adsorbed on lipid and polymeric nanocarriers, for intranasal administration which represent a valid approach to over-come issues related to drug transport across the blood brain barrier (BBB).

Although lipidic and polymeric nanoparticles differ in their physicochemical characteristics, both formulations, as demonstrated in our study and also by other authors (Akel et al., International Journal of Pharmaceutics. 2021) have been envisioned as promising approaches to overcome some limits related to the drug administration into the nasal cavity, such as local enzymatic degradation and mucociliary clearance, providing a method of bypassing the BBB to deliver therapeutic agents to the brain.

Reviewer 2 Report

In this study, lipidic and polymeric nanoparticles were utilized as carriers for a TRAIL-neutralizing monoclonal antibody for the treatment of Alzheimer’s disease (AD). Through intranasal administration of these formulations, the amount of anti-TRAIL antibodies was shown to increase in the brain. The manuscript is written in detail and organized way. The introduction covers enough related literature and the experimental part provides details for the applied procedures. The following issues would be better if addressed properly before publication :

  1. It would be better for comparison if the scale bars of images in Figure 1 are kept the same.
  2. In the text, it is written as "The effective absorption of the anti-TRAIL monoclonal antibody on the NPs surface was evidenced by the reduction in absolute values of ZP for both NANO-A and NANO-B complexes with respect to the unloaded ones, as demonstrated by other authors [35,36]“ for zeta potential values of nanoparticles A and B, fort he loaded ones, compared to non-loaded ones. However, Table 1 shows that antibody loading to PLGA NP has increased the size, which seems to be more relevant with the literature. For lipid-based NANO-B nanoparticles, ZP value seems to be decreasing. The values should be checked, or their sign?
  3. Antibody-loaded NANO-A (543 nm) and NANO-B (243 nm) has very different hydrodynamic volumes, so do authors have any data about their internalization behavior in in vitro conditions?
  4. The authors clearly state the effect of NP size on tissue penetration and their accumulation, „mean size can affect the direct transport through the olfactory or trigeminal axonal nerves. The difference in the mean size evidenced between the two nano complexes deeply influenced their distribution in the hippocampus after in intranasal administration.“ It is understood from the data given at Figure 5C, that smaller NP accumulates in the brain more. However the composition of NP carriers is different, one of them is made out of a hydrophobic PLGA polymer, on the other hand, the other is the self-assembly of lipid molecules stabilized by surfactants. Although their antibody content is similar, their composition, size and even surface charge (the former one is – in total and the latter is +) are different. Do authors have any data about the stability or biodegradability of these NPs in different pH values, or how they have verified the structural integrity of NPs after tissue penetration seems to be unclear or confusing when it comes to a clear comparison for these two nanoparticle delivery systems?
  5. Likewise, is there any antibody release data from these nanoparticle systems?

Author Response

Reviewer #2 Comments and Suggestions for Authors:

In this study, lipidic and polymeric nanoparticles were utilized as carriers for a TRAIL-neutralizing monoclonal antibody for the treatment of Alzheimer’s disease (AD). Through intranasal administration of these formulations, the amount of anti-TRAIL antibodies was shown to increase in the brain. The manuscript is written in detail and organized way. The introduction covers enough related literature and the experimental part provides details for the applied procedures. The following issues would be better if addressed properly before publication:

Q1. It would be better for comparison if the scale bars of images in Figure 1 are kept the same.

R.1) As suggested by the Referee, for the sake of comparison we modified the scale bar making it the same for all figures.

Q2. In the text, it is written as "The effective absorption of the anti-TRAIL monoclonal antibody on the NPs surface was evidenced by the reduction in absolute values of ZP for both NANO-A and NANO-B complexes with respect to the unloaded ones, as demonstrated by other authors [35,36]“ for zeta potential values of nanoparticles A and B, fort he loaded ones, compared to non-loaded ones. However, Table 1 shows that antibody loading to PLGA NP has increased the size, which seems to be more relevant with the literature. For lipid-based NANO-B nanoparticles, ZP value seems to be decreasing. The values should be checked, or their sign?

R.2) Referred to the zeta potential values, the check has been done and the values are corrected. In the text highlighted by the Referee, the authors showed the variations of this parameter in absolute values. In fact, zeta potential increase from -29 to -19 for NANO-A and decrease from +54 to +34 for NANO-B. If we consider the absolute values, both nanosystems evidenced a reduction in their surface properties due the presence of protein (29→19 and 54→34).

Q3. Antibody-loaded NANO-A (543 nm) and NANO-B (243 nm) has very different hydrodynamic volumes, so do authors have any data about their internalization behavior in in vitro conditions?

R.3) In this manuscript we did not report the in vitro internalization data. Literature data and our previous works demonstrated that uptake is strictly dependent by size.  In fact, in Musumeci et al., 2014 and Bonaccorso et al., 2021, the authors demonstrated that the difference in surface charge or in mean size influenced the uptake of NPs. We hope to receive other funding to carry out further investigations.

  1. Bonaccorso A., Pellitteri R, Ruozi B, Puglia C, Santonocito D, Pignatello R, Musumeci T. Curcumin Loaded Polymeric vs. Lipid Nanoparticles: Antioxidant Effect on Normal and Hypoxic Olfactory Ensheathing Cells. Nanomaterials (Basel). 2021; 11(1):159. doi: 10.3390/nano11010159.
  2. Musumeci T, Pellitteri R, Spatuzza M, Puglisi G. Nose-to-Brain Delivery: Evaluation of Polymeric Nanoparticles on Olfactory Ensheathing Cells Uptake. Journal of Pharmaceutical Science. 103(2):628-35. (2014) doi: 10.1002/jps.23836.

Q4. The authors clearly state the effect of NP size on tissue penetration and their accumulation, „mean size can affect the direct transport through the olfactory or trigeminal axonal nerves. The difference in the mean size evidenced between the two nano complexes deeply influenced their distribution in the hippocampus after in intranasal administration.“ It is understood from the data given at Figure 5C, that smaller NP accumulates in the brain more. However the composition of NP carriers is different, one of them is made out of a hydrophobic PLGA polymer, on the other hand, the other is the self-assembly of lipid molecules stabilized by surfactants. Although their antibody content is similar, their composition, size and even surface charge (the former one is – in total and the latter is +) are different. Do authors have any data about the stability or biodegradability of these NPs in different pH values, or how they have verified the structural integrity of NPs after tissue penetration seems to be unclear or confusing when it comes to a clear comparison for these two nanoparticle delivery systems?

Q5. Likewise, is there any antibody release data from these nanoparticle systems?

R.4, R5) We agree with referee, he/she suggests interesting further investigation to deeply develop the project. We will perform stability studies, in vitro internalization and evaluation of efficacy including antibody release, as above indicate, when the project will receive other funding to support its development.

Reviewer 3 Report

This manuscript by Musumeci et al reports an intranasal delivery strategy of a TRAIL neutralizing monoclonal antibody-loaded PLGA nanoparticles which can potentially be used to treat Alzheimer’s disease. The experiments were well designed and the data can support their conclusion. The authors also provide detailed experimental procedures so that others can easily repeat their results. The author also provided in-depth insight into the findings from this study. Thus, I recommend the manuscript for publication after minor revision.

  1. In Figure 2, Figure 3, and Figure 5C, the reviewer suggests the author to add the original data points to these bar graphs so that the readers can appreciate the experimental results of this study.
  2. In Figure 5ab, the review suggests the author to add scale bars to these fluorescence microscope images so that the readers can know the size of these fluorescence signals.
  3. In the Introduction section, when the author mentioned “Moreover, the encapsulation of the drug in the lipid matrix protects the molecule against enzymatic 86 degradation.” The author may also need to cite the reference which shows the same merit of this claim: Nanotheranostics, 2019, 3, 166-178. “The physico-chemical versatility, surface charge, shape, as well as the appropriate size of nanocarriers, are known to affect their ability to overcome biological membranes and the in vivo fate after administration” Reference: NMR in Biomedicine, 2013, 26, 1176-118; Nanoscale, 2022, 14, 4448-4455.

Author Response

Reviewer #3 Comments and Suggestions for Authors:

This manuscript by Musumeci et al reports an intranasal delivery strategy of a TRAIL neutralizing monoclonal antibody-loaded PLGA nanoparticles which can potentially be used to treat Alzheimer’s disease. The experiments were well designed and the data can support their conclusion. The authors also provide detailed experimental procedures so that others can easily repeat their results. The author also provided in-depth insight into the findings from this study. Thus, I recommend the manuscript for publication after minor revision.

Q1. In Figure 2, Figure 3, and Figure 5C, the reviewer suggests the author to add the original data points to these bar graphs so that the readers can appreciate the experimental results of this study.

R.1) Thanks to the Referee for the suggestion. We have now added the original data points to the bar graphs of Figures 2, 3, 5C.

Q2. In Figure 5ab, the review suggests the author to add scale bars to these fluorescence microscope images so that the readers can know the size of these fluorescence signals.

R.2) As suggested by the Referee, now the scale bars for fluorescence images have been added in Figure 5.

Q3. In the Introduction section, when the author mentioned “Moreover, the encapsulation of the drug in the lipid matrix protects the molecule against enzymatic 86 degradation.” The author may also need to cite the reference which shows the same merit of this claim: Nanotheranostics, 2019, 3, 166-178. “The physico-chemical versatility, surface charge, shape, as well as the appropriate size of nanocarriers, are known to affect their ability to overcome biological membranes and the in vivo fate after administration” Reference: NMR in Biomedicine, 2013, 26, 1176-118; Nanoscale, 2022, 14, 4448-4455.

R.3) Thanks to the Referee suggestion. The suggested References have been added in correspondent Introduction section (References 28, 29, 32).

Round 2

Reviewer 1 Report

The authors have revised the manuscript accordingly and can be accepted in the present form after spell check of the manuscript.

Author Response

Thank you, we revised the MS in accordance to suggested changes

Reviewer 2 Report

Compared to the previous version, the newly submitted manuscript seems to have lots of issues addressed with required detailed and clearer explanations. The reverse trend seen for zeta potential values of nano-complexes was mentioned in the discussion part, together with experimental details for the preparation of nano-formulations. Now, the manuscript is more fluidic and understandable. It can be published after a quick check for typo errors like line427 (due to), line403 (the death), line 340 (nanoparticle), line248 (mean) etc. 

Author Response

(The authors gave the same response as above.)
